# Green Energy Optimization in Dinajpur, Bangladesh: A Path to Net Neutrality

Sultana Sharmin, Helmut Yabar *📧 and Delmaria Richards

Graduate School of Science, Technology and Information Sciences, Tsukuba University, 1-1-1, Tennodai, Tsukuba City 305-8577, Japan
* Correspondence: yabar.mostacero.h.ke@u.tsukuba.ac.jp; Tel.: +81-29-853-4269

**Abstract:** Bangladesh has endured a significant power crisis as its economy grows. Hence, it is crucial to investigate the 40% expansion of renewable energy to attain the 2041 renewable energy goal as delineated by the government of Bangladesh. The study explores the current agricultural waste situation in rural areas of the Dinajpur District to propose a feasible alternative and integrated waste management system to meet the energy policy targets for animal waste and crop residues. It analyzed the spatial distribution of feedstocks, identified the optimal sites for the locations of biogas plants based on socioeconomic and environmental criteria and geographic information, and evaluated biogas production to satisfy electricity demand using geographic information system (GIS) suitability analysis and hotspot analysis by proposing six different scenarios. The results show that 2.81 million tons of total agricultural residues are sufficient to produce 11.31 million $m^3$ per year of biogas in the study area. Furthermore, it is found that 21 biogas-based power plants using cattle manure and rice straw are spatially and technically feasible to produce 6389.14 kW of electrical energy per year, which meets 5.73% of the demand of the district in 2019. From the 6 proposed scenarios, number 4 can produce the maximum electricity, 3047.41 kW/year. The findings support the target of achieving a clean, green, sustainable energy system in Bangladesh while improving agricultural residue management. Estimating substrate availability and location is one of the first steps in promoting biogas-based energy from cattle manure and rice straw, which demands comprehensive technical, economic, and social policy reforms. Moreover, bioenergy expansion in Dinajpur District via biogasification represents a commitment to long-term investments in rural areas of Bangladesh.

**Keywords:** biogas; emission reduction; analytical hierarchical process (AHP); recycling rice straw; manure management; renewable energy

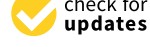

## 1. Introduction

Industrialization, long-term economic prosperity, poverty alleviation, and environmental sustainability in the modern world depend on energy. With the sustainable economic expansion, energy demand has climbed by 7% yearly. However, it is challenging to ensure complete accessibility of the power supply [1]. Only 30% of the population has access to the national grid, and over 75% inhabit rural regions [2]. Electricity demand is anticipated to increase to 34,000 MW by 2030, despite Bangladesh having the lowest per capita energy consumption globally, which amounts to approximately 321 kWh [3]. Natural gas has contributed to 62.2% of the conventional energy sources used to produce electricity in Bangladesh, Figure 1 [4]. On the other hand, natural gas reserves remain limited, with just 15.32 trillion cubic feet accessible at the end of 2020 [5].

According to Abolhosseini et al. in 2014 [6], replacing fossil fuels with renewable energy sources and improving energy efficiency are two significant ways to reduce $CO_2$ emissions and combat climate change. As a result, the government of Bangladesh plans to produce 30% of its total power demand from renewable sources by 2030, but only 1.43% is estimated to be achieved in 2021 [7], Figure 1. The low productivity of renewables has been

attributed to social, economic, and technological barriers. Such social challenges include limited knowledge regarding the benefits of renewable energy among residents suffering from the "Not in my Backyard (NIMBY)" syndrome for renewable energy generation installation due to odor and aesthetics and insufficient land space due to high population density. Additionally, investors feel insecure about investing in renewable energy installation as most energy sources are environment- and weather-dependent. Technologically, there is inadequate knowledge regarding renewable energy infrastructure construction, support, operation procedures, and regular maintenance [8]. This research offers a pathway to aid in meeting the 30% target of 2030 if applied to various districts across Bangladesh since the methodology is malleable with similar feedstock under comparable benchmarks.

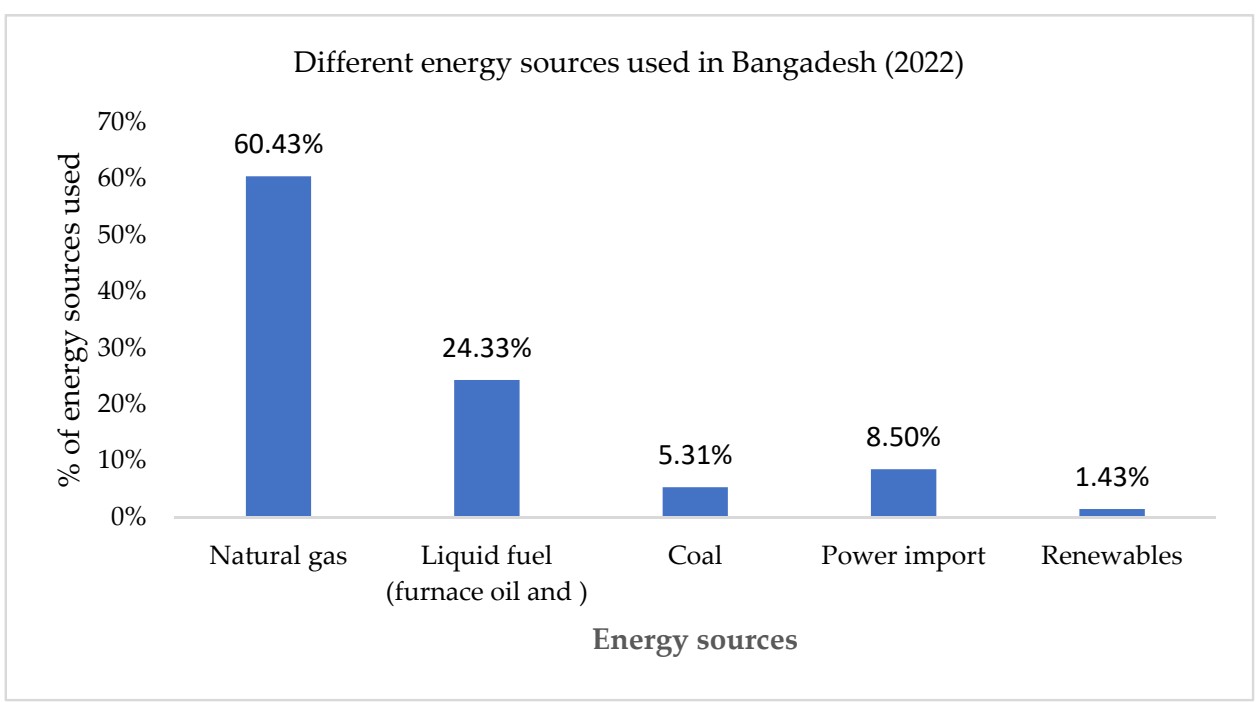

**Figure 1.** Energy source generation in Bangladesh as of 2022. Source: [9].

Despite the imbalance in renewable energy output by source type (Figure 2), there is an immense capacity to improve bioenergy generation by 25 MW by 2020, with biogas accounting for 5% [9]. As reported in the work of Islam et al. in 2014 [10], when compared to other renewable energy-producing sources, including solar, wind, thermal treatment, and hydropower, Bangladesh has immense potential for electricity generation from biomass energy sources such as agricultural crop residues (160.93 terawatt hours), animal waste (121.768 terawatt hours), sawdust, fuel wood, and forest residues (29.91 terawatt hours). Furthermore, due to favorable agro-climatic conditions and hybrid high-yield variety farming, total rice production reached 35.3 million metric tons (MMT) in 2019 [11] and might increase to 47.2 MMT per year by 2050 [12]. However, approximately 70% of the total crop residues consist of rice residues, and the top portion is burned in the field, which causes soil erosion, air quality degradation, and climate change impacts [13].

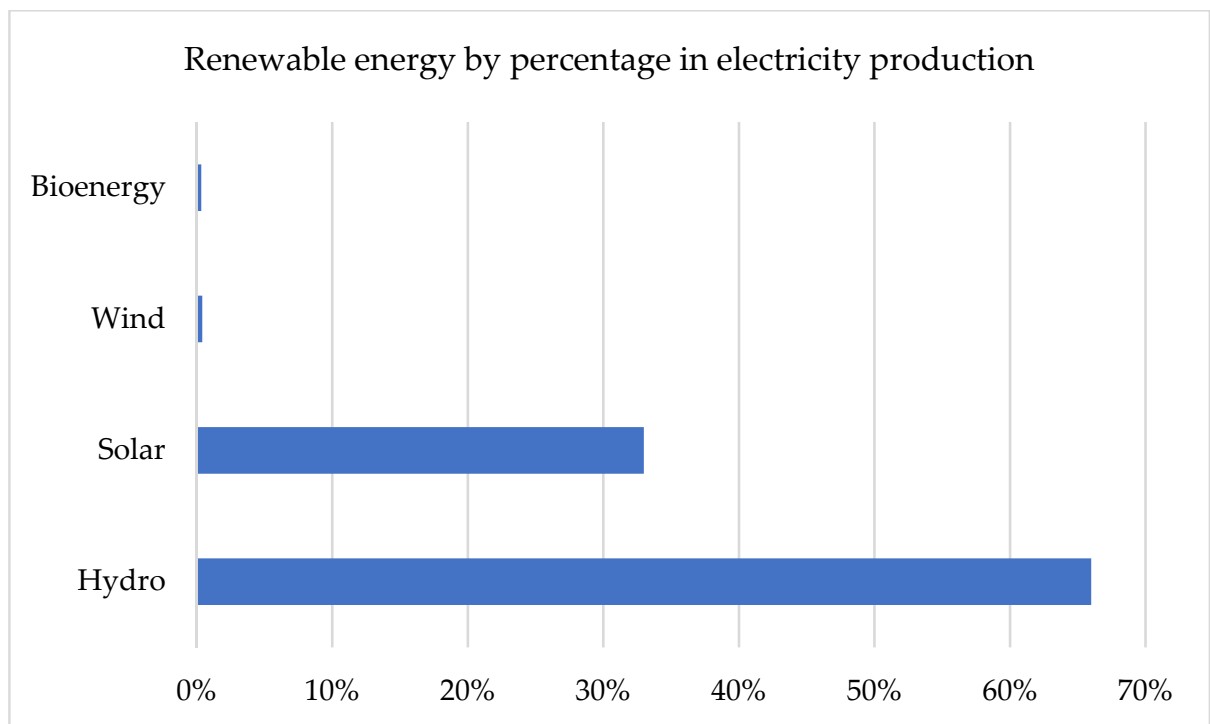

**Figure 2.** Bangladesh's renewable energy generation by energy sources. Source: [9].

Furthermore, crop residue burning in the fields releases massive volumes of carbon monoxide, nitrous oxide, carbon dioxide, methane, volatile hydrocarbons, and particulate matter into the atmosphere, contributing to climate change [14]. About 39% of total greenhouse gas (GHG) emissions have come from agriculture [15] and might continue to be high, with an estimated percentage of total emissions being 28% in 2025 [16]. In addition, Bangladesh produces 155.8 million tons of livestock manure annually, of which a sizable portion is wasted and negatively impacts the environment [17]. By releasing methane (57%) and carbon dioxide (18%) into the atmosphere, manure contributes to the GHG effect, which pollutes the environment [18].

Waste-to-energy initiatives from anaerobic co-digestion of agricultural residues and animal waste are crucial options for reducing reliance on fossil fuels while simultaneously lowering GHG emissions from improper agricultural waste management. Within the renewable energy mix of Bangladesh, there were targets to reach 30.4 MW and 7.28 MW electricity production capacity from biomass of agricultural residue and biogas from municipal waste, respectively, by 2021 [19]. However, the outcome is unconfirmed. Nonetheless, evidence has shown that with the financial and technical support of the Netherlands Development Corporation and the World Bank, the Bangaladesh Infrastructure Development Company Limited (IDCOL) has built 48,000 biogas plants in different districts of the country [20]. However, although biogas from anaerobic digestion is an innovative technology, Bangladesh's energy production from this source is insufficient, accounting for only 0.68 percent of total power generation [21]. Consequently, it is essential to investigate appropriate measures to enhance biogas production utilization for approved regulatory services.

A geographic information system (GIS) is a computer system that examines and demonstrates geographically referenced information (The United States Geological Survey) [22]. In addition to being a vital tool for addressing some of the environmental issues associated with the development of biowaste, the planning of farm waste recycling programs in optimal sites at optimal capacities is both practical and financially feasible. Due to the significant geographical dispersivity of farms, using spatial information technologies like remote sensing and GIS to address the location of bioenergy plants seems to be a

beneficial practice. Various studies have used GIS as a suitable tool for site appropriateness, indicating its capacity to address location-related problems. Consequently, this study takes advantage of the opportunities for effective bio-resource mapping. Therefore, the purpose of this study was to establish a logical framework that would act as a road map for choosing appropriate locations for biogas plants using the power of geospatial technology [23].

It employs data attached to a unique location to deepen understanding of phenomena in particular environments. Moreover, GIS has proven to be a valuable tool in biomass energy research for analyzing and evaluating renewable energy resources since it points to suitable locations for biogas plants and enables the determination of the most economically viable use of available feedstocks [24]. Previous research has focused on agricultural waste utilization for bioenergy by employing a GIS-based analytical hierarchical process (AHP), an integrated decision-making tool. These include the use of rice straws and animal dung as viable feedstocks.

Perpina et al. (2013) [25] and Silva et al. (2014) [26] identified AHP as an effective method for finding suitable locations for biomass plants. Furthermore, a study by Sahoo et al. in 2018 [27] showed the location of biogas plants from available crop residues through a GIS-based model with multi-criteria analysis. Additionally, Venier and Yabar, 2017 [28] studied optimal sites for biogas plants by using spatial statistical GIS-based suitability analysis, considering geographical, environmental, and socioeconomic criteria in Buenos Aires Province. Yet, there are few details on Bangladesh's appraisal of rural waste as a substantial source of bioenergy generation.

Notably, the work of Akther et al., 2019 [29] focuses on site suitability analysis of biogas digester plants for municipal waste using GIS and multi-criteria analysis for waste management as a potentially effective option to produce bioenergy from organic waste in Dhaka City. It selects the most suitable sites for biogas digesters with a multi-criteria analysis focusing on four factors environmental, social, safety, and economics. Ahmad and Wu, 2022 [30] examined household-based factors affecting the uptake of biogas plants in Bangladesh and their implications for sustainable development. Using logistic regression analysis, the authors included 262 biogas consumers and 312 non-consumers for the Dhaka Division (Bangladesh) in the research sample. The lack of spatial identification of agricultural manure and rice straw wastes and their potential contribution to uptake in biogas generation and subsequent utilization warrant the noteworthiness of this study.

This study's primary goal is to assess the existing agricultural waste availability in the rural part of Dinajpur, Bangladesh, which is the study area, and propose a feasible alternative integrated waste management system to aid in achieving the energy policy targets of 2009 [31]. Therefore, the specific objectives of the study are to: (1) utilize GIS to ascertain suitable locations for biogas plant installation where energy from livestock waste (cattle manure) and crop waste (rice straw) are produced; and (2) assess the potential of agricultural waste to energy using qualitative, quantitative, and practical methodologies in proposed scenarios suitable for the study area. Additionally, the study highlights the probability of GHG emission reduction from the co-fermentation of agricultural waste, which has not been considered in past studies. Thus, this research bolsters it as a unique point. Moreover, it emphasizes how improved manure management can positively impact sustainable farming in the Dinajpur District, another limitation of past integrated livestock manure management studies. Evidence gathered from the scrutiny of past studies indicates an inept understanding of resource recovery in the form of electricity for fossil fuel displacement from agri-waste is warranted. Thus, it is hypothesized that the available agri-waste can generate a viable amount of biogas for electricity after reduction by energy conversion factors.

*The Role of Anaerobic Fermentation in the Biogas Process*

Anaerobic digestion (AD) is a precisely balanced ecological environment involving the subsequent breakdown of biodegradable macromolecules, including carbohydrates, proteins, and lipids, into soluble organics for producing biogas by diverse groups of bacteria

and archaea without the presence of molecular oxygen. As a result, AD produces biogas, a beneficial gas that can be converted into electricity. Biogas mainly comprises 30–40 percent carbon dioxide, 50–70 percent methane, and trace amounts of other gases [32]. The use of organic waste materials in the AD process means it offers a new platform for waste treatment, nutrient recovery, and bioenergy production, directly and indirectly contributing to reducing $CO_2$ emissions. The proportions of the gases depend on the raw materials and other process parameters like the hydraulic residence time (HRT) and temperature. The energy content of biogas is about 60% (depending on the methane content) compared to natural gas [33] and can be easily adapted for use as a replacement for natural gas. In biogas production, the sequence of microbial activities that occur is conceptually divided into four stages: hydrolysis and fermentation, acidogenesis, acetogenesis, and methanogenesis [34].

## 2. Materials and Methods

### 2.1. Study Area

Dinajpur District is situated at 25°14′–25°38′ N and 88°05′–88°28′ E and is bordered by Thakurgaon (North), Panchagarh District (East), Nilphamari and Rangpur districts (South) and Gaibandha and Joypurhat districts (Southwest by the West Bengal of India). The district consists of thirteen subdistricts, eight municipalities, and one hundred and one unions. The administrative boundary of Dinajpur District is illustrated in Figure 3.

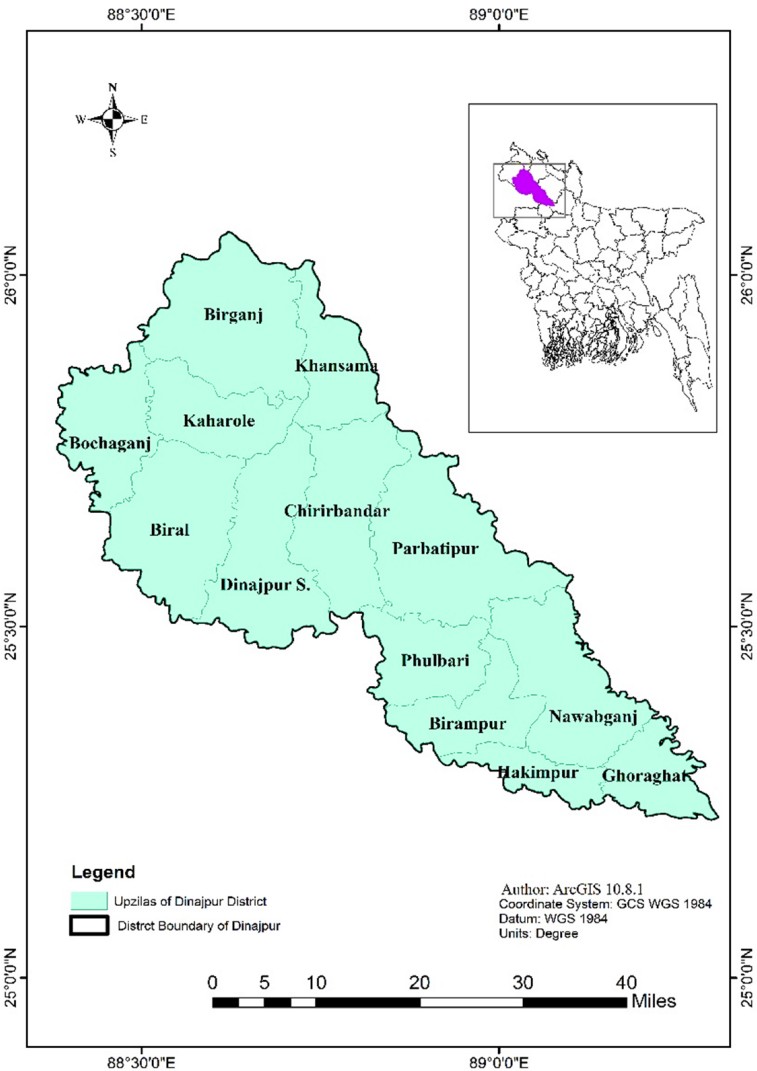

**Figure 3.** Study area—Dinajpur District of Bangladesh.

About 42.85% of people's main occupation is agriculture, and the total cultivable land area is 288,432 hectares [35]. Dinajpur District has the third largest cattle population in the country, consisting of 860,595 in 2019 [36], and occupies the second position among all districts of Bangladesh with 1,472,046 metric tons of rice in 2020 [37]. Therefore, improper agricultural waste management adversely affects residents' public health, the natural environment, and the country's economic growth. Agricultural wastes in the study represent organic wastes from crops and livestock manure. If appropriately treated by biogasification or composting, this amount of waste, including rice straw and cattle manure, can be a valuable renewable energy source, primarily through anaerobic co-digestion.

*2.2. Methods*

The study is conducted to determine the potential biogas production sites using cattle manure and rice straw as substrates by analyzing geographic, environmental, and socioeconomic criteria. It has been divided into theoretical assessment, site suitability, and spatial statistics analyses. The theoretical analysis examines the availability of livestock and crop residues based on mathematical methods. In the site suitability analysis, different criteria are considered for creating a restriction map, focusing on socioeconomic and environmental factors with AHP techniques utilized for suitability mapping. The final suitability map of potential areas for sitting biogas plants is made by combining a restriction map and a suitability map. The detailed research methodology framework is shown in the following Figure 4.

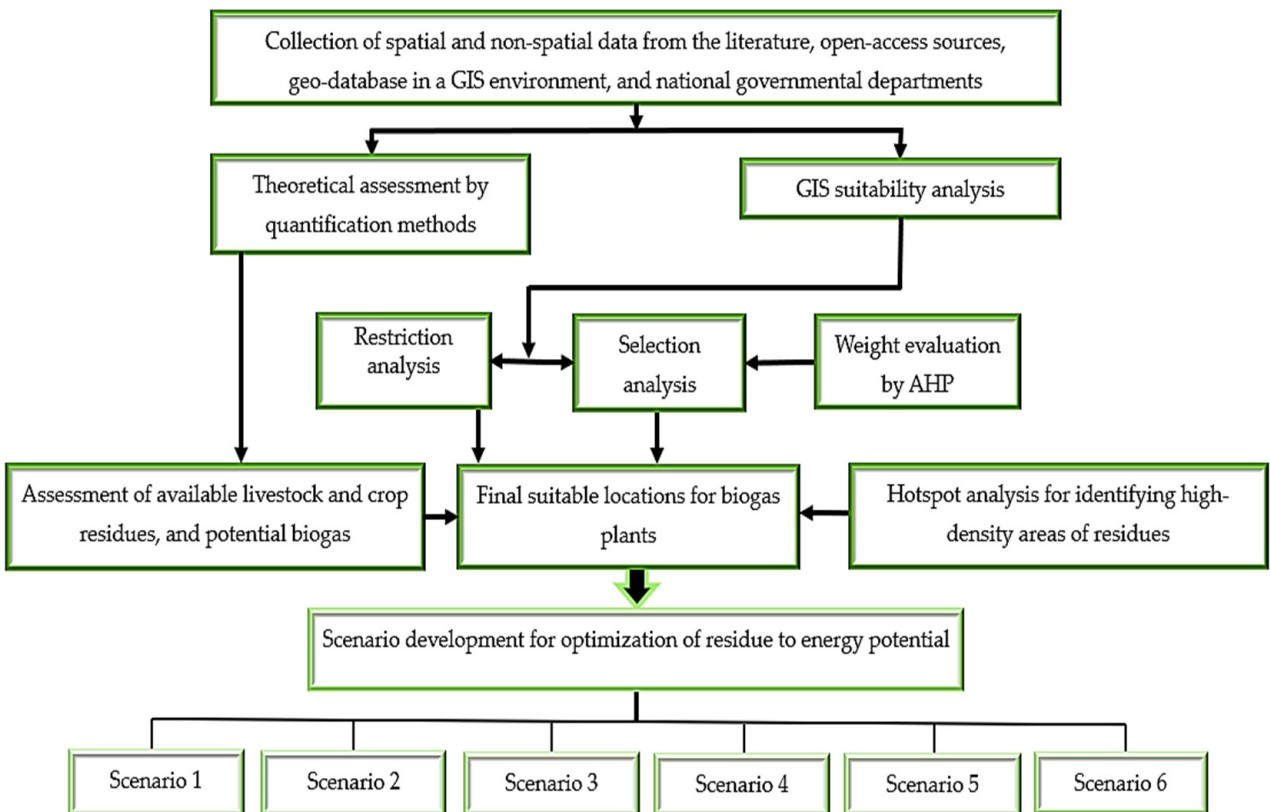

**Figure 4.** The research framework.

In the spatial statistics analysis, high-density areas of feedstocks are estimated through hotspot analysis, and some scenarios are designed contingent on feedstock intensity within significant areas. ArcGIS software version 10.8.1, created by the Environmental Systems Research Institute in New York City, United States of America, was used for estimating the location, scope, size, and capacity of energy output from biogas plants under proposed scenarios.

### 2.2.1. Quantification of Potential Energy from Cattle Manure and Rice Straw

All agricultural data used in the analyses in the study were collected from the annual reports of the Department of Agricultural Extension (DAE) in 2019 [38] and the Department of Livestock Services (DLS) in 2021 [36]. The theoretical model has been modified from Rahman and Paatero, 2012 [39] and Suhartini et al., 2019 [40] for estimating the total agricultural residue amount and biogas potential from the residues of the study area. The relevant equations are depicted below.

$$C_M = C_P \times \text{unit factor} \left( \text{kg head}^{-1} \text{day}^{-1} \right) \times 365 \tag{1}$$

where $C_P$ is the head counts in thousands, and the average fresh manure productions for cattle were 2.86 kg head$^{-1}$ day$^{-1}$ dry matter [41]. This study assumes 35% electrical conversion efficiency for calculating potential power capacity over 300 days of annual operation, as in the work of Richards and Yabar, 2022 [42] and Suhartini et al., 2019 [40]. Theoretical biogas potential $\text{TPB}_{(CM)}$ (m$^3$/year) of cattle manure is calculated by Equation (2).

$$\text{TPB}_{(CM)} = C_M \times A_C \times VS_M \times B_{CM} \tag{2}$$

where, $A_C$ is the available coefficient for cattle manure, $VS_M$ is the ratio of volatile solids to dry matter in cattle manure and $B_{CM}$ is the biogas yield in m$^3$ kg$^{-1}$ TS 60% methane content (MC) is assumed in biogas for large animal manure [43]. Methane emissions (m$^3$ $CH_4$) from the biogas production of cattle manure can be shown in Equation (3).

$$CH_{4(CM)} = \text{TPB} \times MC_{(CM)} \tag{3}$$

The energy value of methane from biogas ($E_{CH_4}$) is 10.5 kWh/ m$^3$ [44]. As a result, Equation (4) can be used to compute the total energy potential of cattle dung from the methane produced for biogas $E_{CM}$ (kWh y$^{-1}$).

$$E_{CM} = CH_{4(CM)} \times E_{CH_4} \tag{4}$$

The surplus availability factor is vital for obtaining available residues, as a small amount of straw is left in the field for fertilizer but the majority is used for cattle and fish feed [45]. Therefore, to estimate the total amount of total rice straw generation (Ton y$^{-1}$) in the study area, Equation (5) below is used:

$$RS_R = R_P \times RPR \tag{5}$$

where, $R_P$ is the rice production in metric tons and RPR is the residue-to-crop yield ratio. Many studies show that the rice straw production residue-to-crop yield mass ratio is between 1.5 and 1.76 [46,47]. Thus, 1.76 is used in this study. Available crop residues for biogas production (ton y$^{-1}$) are calculated by Equation (6).

$$A_{CR} = RS_R \times RS_{CF} \times RS_{SAF} \tag{6}$$

$RS_{CF}$ stands for residue collection factor (kg kg$^{-1}$ of residue) and $RS_{SAF}$ stands for surplus accessibility factor (kg kg$^{-1}$) of residue. Theoretical biogas potential, $\text{TPB}_{(RS)}$ (m$^3$/year) of available crop residue for biogas is calculated by Equation (7).

$$\text{TPB}_{(RS)} = A_{CR} \times RS_{DF} \times V_C \times B_{RS} \tag{7}$$

where, $RS_{DF}$ is the residue dryness factor (kg kg$^{-1}$ of residue), $V_C$ is the proportion of volatile solid (VS) to dry matter (DM), and $B_{RS}$ is the biogas generation rate from volatile solids (m$^3$ kg$^{-1}$ of VS) of rice straw. It is assumed that 0.226 m$^3$ kg VS$^{-1}$ methane yield

$MC_{(RS)}$ in the biogas for rice straw [48]. Methane emissions, $CH_{4(RS)}$ (m³ CH₄) from the biogas production of rice, straw are generated by Equation (8).

$$CH_{4(RS)} = TPB_{(RS)} \times MC_{(RS)} \tag{8}$$

As a result, the total energy potential from methane produced by the biogas of rice straw $E_{RS}$ (kWh y $^{-1}$) is calculated by Equation (9).

$$E_{RS} = CH_{4(RS)} \times E_{CH_4} \tag{9}$$

Several factors of all residues plus methane potentials, including the collection factor, the ratio of volatile solids to dry matter, and the surplus accessibility factor, are derived from renowned literature, as outlined in Tables 1 and 2. The study used anaerobic digestion (AD) for the biogasification of manure on farms with more than 50 animals per head to calculate the potential energy.

**Table 1.** Parameters used to evaluate overall availability for livestock residue.

| Livestock Residue | Residue (Dry Matter) Generation Rate | Collection Factor of Manure | The Ratio of Volatile Solids to Dry Matter | Biogas Generation Rate |
|---|---|---|---|---|
| Unit | kg dry matter/day | 0–1 | 0–1 | m³/kg VS |
| Cattle manure | 2.86 | 0.5 | 0.93 | 0.66 |
| Reference | [41] | [42] | [41] | [49] |

Source: Detailed values are adopted from the listed references in the table.

**Table 2.** Parameters used to evaluate overall availability for agricultural residue.

| Crop Residue | Residue Collection Factor | Surplus Accessibility Factor | Residue Dryness Factor | Volatile Solids to Dry Matter Proportion | Biogas Generation Rate from Volatile Solids |
|---|---|---|---|---|---|
| Unit | 0–1 | 0–1 | 0–1 | 0–1 | m³/kg VS |
| Rice straw | 0.6 | 0.8 | 0.87 | 0.54 | 0.34 |
| Reference | [41] | [41] | [41] | [42] | [42] |

Source: Detailed values are adopted from the listed references in the table.

### 2.2.2. Criteria for Restriction

The following steps were performed for the restriction analysis: The first steps after selecting restriction criteria were to add six feature files, including roads, land use, waterways, protected areas, transmission lines, and power plants, to the buffer analysis. In the second step, the geoprocessing tool converted all feature files into rasters for the raster analysis. The third step involves converting the intermediate map layers into a Boolean map using unsuitable (0) and suitable (1) areas in the constraint maps [50]. Finally, the restriction map was created by combining all restriction factors using a raster calculator. The buffer criteria for numerous restriction factors used in this study are based on secondary publication data from previous studies of biogas, solar, and other renewable energy plants, as shown in Table 3, to create an appropriate distance for making buffers for all layers discussed in the restriction analysis.

**Table 3.** Buffering criteria for restriction mapping in meters.

| References | Built-Up Areas (Residential Houses, Commercial Buildings, Cemetery, Farms, and Recreational Places) | Transport (Central and Local Roads Distance) | Water Bodies (River, Canal, and Lake) | Protected Areas (National Park and Forest) | Distance from Power Plants and Substations | Distance from Transmission Lines |
|---|---|---|---|---|---|---|
| [51] | 300 | 100 | 200 | - | - | - |
| [52] | 1000 | 30 | 100 | 500 | 200 | - |
| [53] | 500 | - | 200 | - | - | - |
| [54] | 200 | 70 | 150 | - | - | - |
| [25] | 600 | 100 | 500 | 500 | - | - |
| [55] | 1000 | 30 | 200 | 500 | 100 | - |
| [56] | 500 | 150–500 | 300–500 | 1000 | - | - |
| [27] | 1000 | 100 | 150 | 1000 | - | - |
| [26] | - | - | - | - | - | 200 |
| Criteria for this study | 500 | 30 | 100 | 100 | 100 | 100 |

Source: Developed by authors based on references included in the table.

### 2.2.3. Criteria for Suitability

Three criteria, distance from roadways, rivers, and elevations, were chosen from among three elements identified by experts from the literature review of similar studies on environmental, economic, social, and safety issues. First, the study area is susceptible to flooding. Rivers in the area are flooded during the rainy season. Therefore, to reduce the risk of inundation, a distance equal to or greater than 492 m is assigned the lowest risk, and choosing a site is only permitted if it is less than 6.64 m. Second, for road network distance, the theme of "the closer, the better" was followed for reducing collection costs and transportation distances of available resources. It has been proposed that utilizing existing roads is preferable to avoid the costs of new road development [57]. Third, biogas facilities should not be built at higher elevations or in low-lying areas. After all, gathering feedstock, shipping, and grid network connectivity might be complex. Finally, in terms of elevation, a lower elevation is the preferable choice for launching a cost-effective and efficient residue collection approach.

AHP is a robust and comprehensive approach that allows organizations and individuals to consider qualitative and quantitative variables in multiple-criterion decision-making [58]. AHP consists of four primary steps: (1) breakdown of issues into subproblems; (2) pairwise comparison of items; (3) consistency evaluation of paired criteria; and (4) synthesis of the findings to arrive at a final ranking [59]. As a part of the first step, three criteria for the suitability model were already selected as environmental, economic, and social paradigms to solve the multi-criteria evaluation problem of the suitability model in this study. The second step involves utilizing the pairwise comparison of the three evaluation criteria, road network, river, and elevation, which are considered alternatives for the selectivity model. The third and fourth steps are taken subsequently. Finally, a basic numeric scale consisting of the numbers 1–9 and their multiplicative reciprocals in AHP identified in Table 4 creates a pairwise matrix comparison of the priorities according to implicit individual preferences and judgments [60].

**Table 4.** The fundamental numeric scale in AHP.

| Definition | The Intensity of Importance on an Absolute Scale |
| --- | --- |
| Equal importance | 1 |
| Moderate importance | 3 |
| Strong importance | 5 |
| Very strong importance | 7 |
| Extreme importance | 9 |
| Intermediate values between the two adjacent judgments | 2,4,6,8 |

Source: Modified by the author based on Saaty's work (1990) [60].

In this study, a $3 \times 3$ matrix was designed for three criteria: environment, social, and economic, as well as three sub-criteria: road transportation, elevation, and river, based on earlier studies.

$$C = \begin{vmatrix} C_{11} & C_{12} & C_{13} \\ C_{21} & C_{22} & C_{23} \\ C_{31} & C_{32} & C_{33} \end{vmatrix} \tag{10}$$

where, $C_{11} = C_{22} = C_{33} = 1$ and $C_{21} = \frac{1}{C_{12}}, C_{31} = \frac{1}{C_{13}}, C_{32} = \frac{1}{C_{23}}$.

Next, the sum for each column of the pairwise matrices was calculated as follows:

$$C_{ij} = \sum_{i=1}^{n} C_{ij} \tag{11}$$

Each element of the matrix is divided by its column total to synthesize a normalized pairwise matrix seen below:

$$X_{ij} = \frac{C_{ij}}{\sum_{i=1}^{n} C_{ij}} = \begin{vmatrix} X_{11} & X_{12} & X_{13} \\ X_{21} & X_{22} & X_{23} \\ X_{31} & X_{32} & X_{33} \end{vmatrix} \tag{12}$$

A weighted matrix of priorities is produced by dividing each entry in a column by the normalized matrix's column sum, represented in Equation (13):

$$Y_{ij} = \frac{\sum_{j=1}^{n} x_{ij}}{n} = \begin{vmatrix} W_{11} \\ W_{12} \\ W_{13} \end{vmatrix} \tag{13}$$

Next, the consistency ratio is calculated in the equation following three steps. First, the consistency for one criterion is measured in Equation (14):

$$\begin{vmatrix} C_{11} & C_{12} & C_{13} \\ C_{21} & C_{22} & C_{23} \\ C_{31} & C_{32} & C_{33} \end{vmatrix} \times \begin{vmatrix} W_{11} \\ W_{12} \\ W_{13} \end{vmatrix} = \begin{vmatrix} Z_1 \\ Z_2 \\ Z_3 \end{vmatrix} \tag{14}$$

The principal eigenvector ($\lambda$max) was computed as follows:

$$\lambda_{max} = \sum_{i=1}^{n} CZ_{ij} \tag{15}$$

Then, the consistency of the index (CI) is calculated by Equation (16):

$$CI = \frac{\lambda_{max} - n}{n} \tag{16}$$

where x is the most significant eigenvalue of the matrix and n is the number of factors. The consistency judgment must be checked by a consistency ratio (CR) less than 0.1, which is acceptable [59]. CR calculated by Equation (17):

$$CR = \frac{CI}{RI} \tag{17}$$

With n equal to three in this study and the random index (RI) equal to 0.58 obtained from the reciprocal matrices generated randomly using Saaty's scale values shown in Table 5. After obtaining the weight of each criterion from the AHP, the reclassified all raster data, which were classified based on their suitability level, were then put in the weighted overlay process in the spatial analyst tool and multiplied by the associated weights for each criterion using ArcMap's Reclassify Tool for representing the best, good, not good, bad, and worst areas of biogas plant development.

**Table 5.** Value of the random index (RI).

| n | 1 | 2 | 3 | 4 | 5 | 6 | 7 | 8 | 9 | 10 |
|---|---|---|---|---|---|---|---|---|---|----|
| RI | 0.00 | 0.00 | 0.58 | 0.9 | 1.12 | 1.24 | 1.32 | 1.41 | 1.46 | 1.49 |

Source: [61].

### 2.2.4. Suitability Analysis

The constraint and suitability maps were integrated with spatial analysis to discover the best locations for installing biogas plants from different social, environmental, and economic layers. Suitability analysis is determined by Equation (18), as below [62]:

$$S = \sum_{i=1}^{n} W_i C_i \cdot \prod_{j=1}^{m} R_j \tag{18}$$

where S represents suitable sites for digester plants, $W_i$ depicts the weight of the criterion, and $C_i$ means criteria for site selection and facility location. $R_j$ delineates the restriction area, and j stands for the restriction criteria, which include roads; land use (residential homes), electric stations, bus stations, bridges, and pump stations; waterways (lakes, canals, and rivers); public buildings; conservative areas (national parks); transmission lines; and power plants.

### 2.2.5. Hotspot Analysis

Within hotspot analysis (GETIS-ORD GI*), the larger z-score or statistically significant positive z-score indicates the high values (hot spot) of clusters, and the negative values of the z-score show the low values (cold spot) of clusters. The Getis-Ord local statistics are given in Equations (19)–(21) [63]:

$$G = \frac{\sum_{j=1}^{n} w_{ij} x_j - \dot{X} \sum_{j=1}^{n} w_{ij}}{s \sqrt{\frac{\left[ n \sum_{j=1}^{n} w^2_{ij} - \left( \sum_{j=1}^{n} w_{ij} \right)^2 \right]}{n-1}}} \tag{19}$$

where $x_j$ stands for the attribute value for j,
$\quad$ $w_{i,j}$ is the spatial weight between features i and j,
$\quad$ n is equal to the total number of features represented in Equation (20):

$$\dot{X} = \frac{\sum_{j=1}^{n} x_j}{n} \tag{20}$$

$$S = \sqrt{\frac{\sum_{j=1}^{n} x^2_{ij}}{n} - \left( \dot{X} \right)^2} \tag{21}$$

After making hotspot analysis, a map of spatially statistically significant clusters with high potential biogas generation sites is intersected with the final suitability map to determine the best location of higher residue generation sites for identifying the optimal location, number, and size of plants.

### 2.3. Anaerobic Digestion of Agri-Waste for GHG Emission Mitigation

The potential global warming mitigation (PGWM) of biogas production from co-fermentation of cow excreta and rice straw substrates was computed considering greenhouse gas abatement probability via cow manure and rice straw management, that is, (1) feedstock emission factor, (2) feedstock availability, (3) methane emissions $CH_4$, (4) nitrous oxide emissions, and (5) global warming potential (GWP). The first and second revisions of the 1996 and 2019 Intergovernmental Panel on Climate Change Guidelines for National Greenhouse Gas Inventories (IPCC Guidelines) are utilized as references for the computation of GHG emissions from animal cattle manure since they provide a general guide to estimating methane emissions from livestock manure. Tier 1 estimation methodology is utilized with default values from past studies applied relative to the lack of availability in Dinajpur, Bangladesh [64]. Likewise, default emission values for nitrous oxide and methane are adopted from the study of Noorollahi et al. in 2015 [33] and utilized in the computation of emissions from burnt rice straw. The scaled emissions amount is based on potential methane ($CH_4$) and nitrous oxide ($N_4O$) emissions since open field burning of rice straw emits both gases [33]. Consequently, the PGWM was derived by applying Equation (22).

$$\text{PGWM} = \text{feedstock emissions factor (FEF)} \times \text{global warming potential (GWP)} \quad (22)$$

where methane in cattle manure is equivalent to the emission factor of cattle (1.7) multiplied by the total amount of cattle divided by 1,000,000 times the GWP of methane (28), subsequently, GHG emissions from rice straw combustion are equivalent to the emissions factor of rice straw (4.51 g kg$^{-1}$ DW) times the total rice straw burned per year multiplied by the GWP of methane and nitrous oxide (265) [64]. Thereafter, the total emissions reduction is equivalent to the total methane from cattle manure and rice straw. Lastly, the avoided emissions from biogas generation are found by multiplying the amount of biogas from each scenario times Bangladesh's grid emissions factor (0.691) [65]. 1 m3 of biogas is equivalent to 0.93 m$^3$ of natural gas, 0.40 kilogram of furnace oil, 0.52 liter of diesel oil, 0.62 liter of kerosene oil, and 1.46 kilogram of coal which is shown in Table 6.

**Table 6.** The equivalent quantity of different fossil fuel types comparable to 1 m$^3$ of biogas.

| Fossil Fuel Types | 1 m$^3$ of Biogas Equivalent | Source |
|---|---|---|
| Natural gas | 0.93 m$^3$ | [66] |
| Furnace oil | 0.40 kg | [67] |
| Diesel oil | 0.52 L | [68] |
| Kerosene oil | 0.62 L | [68] |
| Coal | 1.46 kg | [66] |

## 3. Results

### 3.1. Estimation of Residue in the Study Area

The total agricultural residue is estimated at 2.815 million tons per year, whereas cattle manure and rice straw account for 0.036 and 2.779 million tons per year, respectively. The available cattle manure and rice residues for biogas production are 0.018 and 1.33 million tons per year, respectively. Therefore, the total biogas potential from the estimated available residue is 11,317,288.94 m$^3$ per year, including 11,103,484.3148 and 213,804.63 m$^3$ per year of biogas from cattle manure and rice straw. Therefore, it is estimated that cattle manure and rice straw generate 69,951,951.18 and 507,358.39 kWh per year, respectively. Additionally, jointly, both residues produce 70,459,309.57 kWh per year.

### 3.2. Restriction Mapping

Different GIS shapefiles are used to exclude unwanted areas to illuminate unwanted areas from the final suitable map. The final restriction map was developed by multiplying each restriction map separately in the raster calculator of ArcGIS 10.8.1 and combining all layers to obtain the final restriction map illustrated in Figure 5.

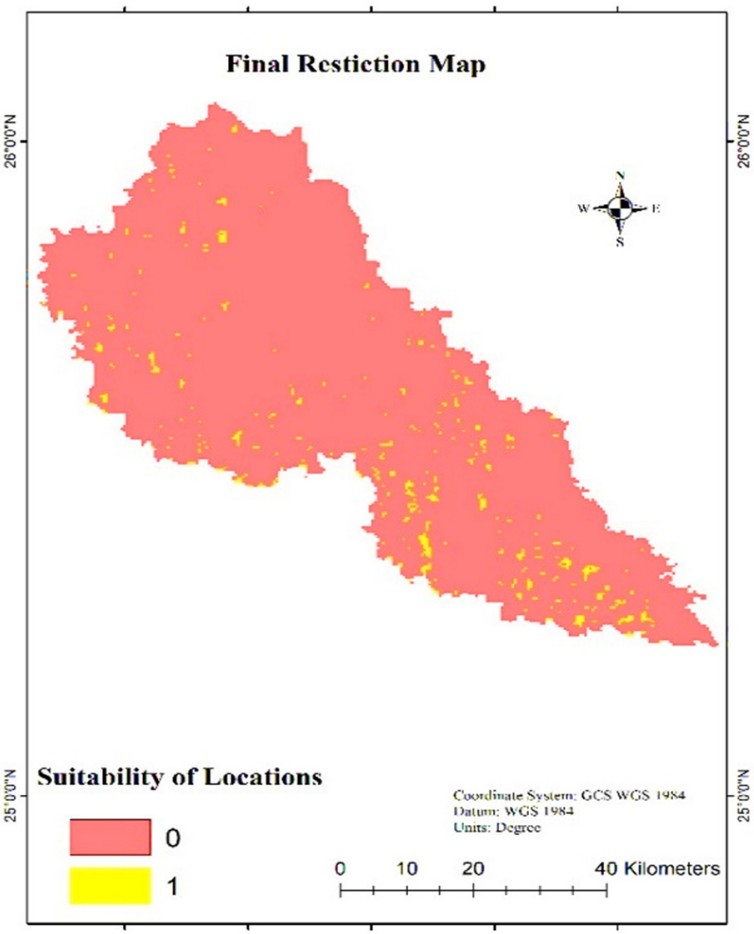

**Figure 5.** Restriction map of the study area.

### 3.3. Weight by AHP Process

Analytic Hierarchy Process is used to obtain weighted preferences based on three criteria: distance from rivers and roads plus elevation. A similar weight matrix is created for pairwise preferences based on economic, environmental, and social safety, as outlined in Table 7.

**Table 7.** Weight preferences for suitability analysis.

| Criteria | Economic | Environment | Social Safety | Weight Preferences | Final Priorities |
|----------|----------|-------------|---------------|--------------------|------------------|
| Road | 0.697 | 0.598 | 0.688 | 0.623 | 0.687 |
| River | 0.232 | 0.120 | 0.235 | 0.066 | 0.225 |
| Elevation | 0.072 | 0.283 | 0.077 | 0.311 | 0.087 |

Source: Calculated by the authors.

All weights above are given based on past literature. Finally, four matrices and the final weighted preferences are used in ArcGIS 10.8.1 software for applying the weighted

overlay tool for designing the suitability maps. The highest priority is found in roads (0.687), followed by the river (0.225), and then elevation (0.087).

### 3.4. Suitability Mapping

The suitability map has been designed by analyzing the suitability of three factors: roads, rivers, and elevations. After combining them, the lands are modeled with three levels (the worst, moderate, and best) for the less suitable and the best areas for developing biogas plants. The most suitable areas are found in the southern parts, and the most moderate areas are found in the northern areas of the district shown in Figure 6.

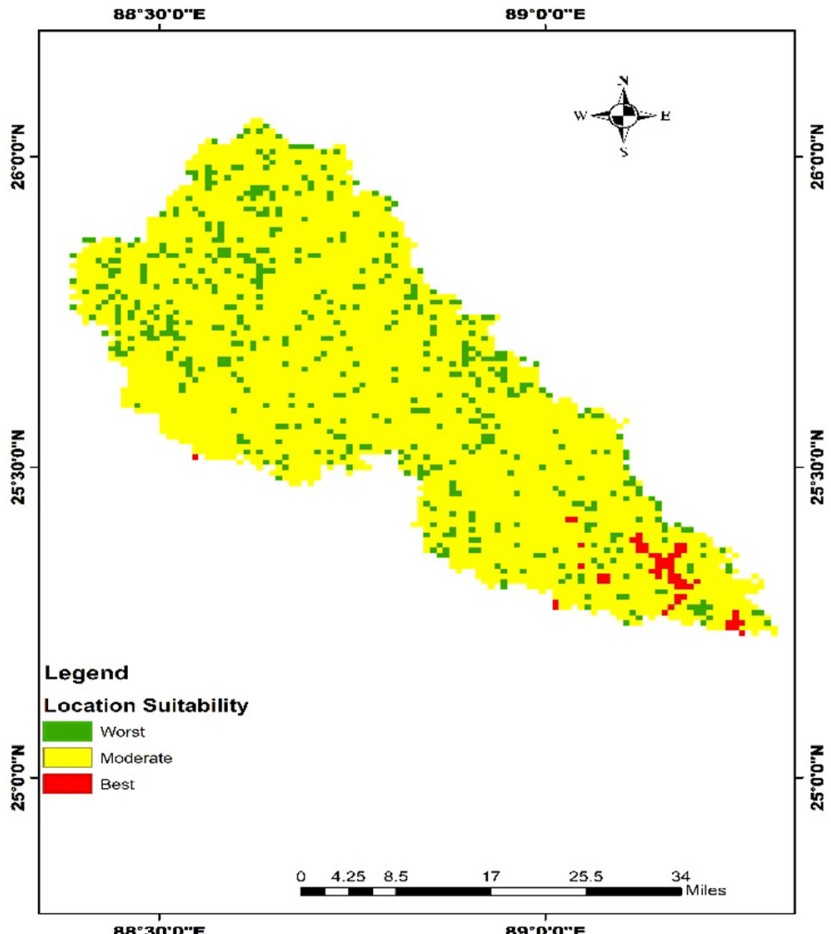

**Figure 6.** Suitability map.

### 3.5. Final Mapping

After combining restriction and suitability maps, the final suitability map was made by the Times Tool in the Arc Tool Box of ArcGIS 10.8.1. The districts consider the best and most moderately suitable areas for sitting biogas plants, whereas less suitable areas are excluded. The suitable areas are generated as raster images, then converted to polygon features and classified with geo-codes for each zone for easy identification, as shown in Figure 7. With the methods above, the final suitability map is used to design six scenarios and appropriate suitable site locations.

### 3.6. Designing Scenarios

By selecting Gi-Bin values of 1, 2, and 3 in hotspot analyses, a map of optimal locations with high-density feedstocks with 99%, 95%, and 90% confidence is shown in Figure 8. This step helped identify suitable sites with the highest electricity potential for installing biogas plants.

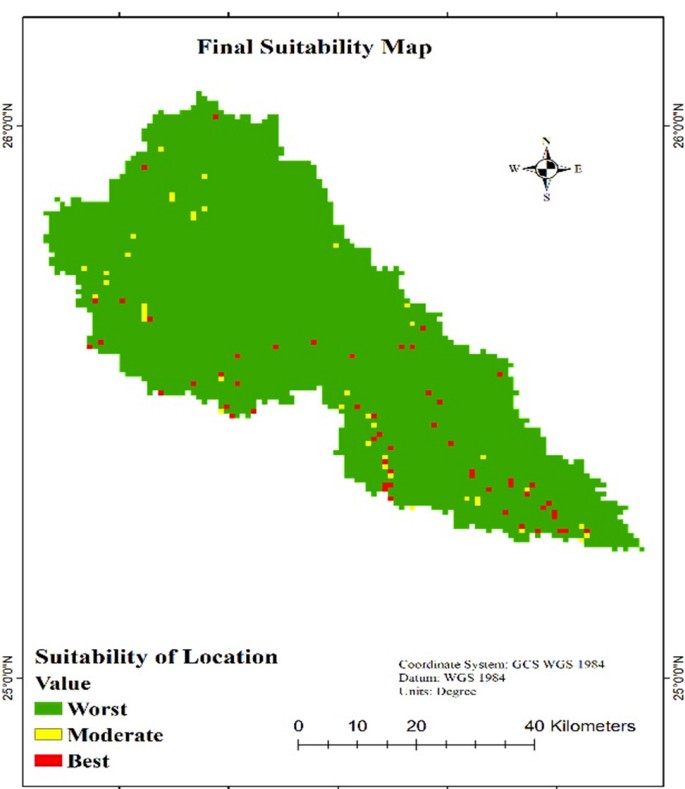

**Figure 7.** Final suitability map.

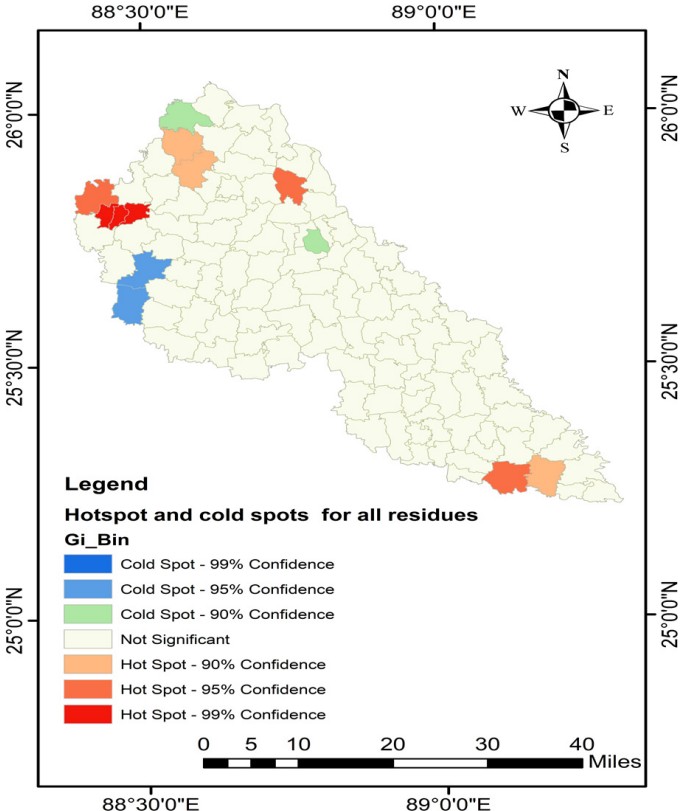

**Figure 8.** Hotspot analysis of available residues in Dinajpur District.

In this research, six scenarios were designed based on the availability of both feedstocks for energy production. Biogas facilities with a power output of 250 kW or more are adopted as a viable investment option [69].

### 3.6.1. Scenario 1

After applying intersection analysis and computations, three biogas plants were seen as viable in the Hakimpur Upazila with a 926.54 kW/year capacity, which satisfied 0.83 percent of Dinajpur District's local electricity demand in 2019 [70]. Therefore, 2019 is utilized as the base year for the compilation of scenarios. For scenario 1, the power generation capacity was 19,884.67 kWh/year, requiring 46,207.71 tons/year of total feedstock. Three potential biogas plants for scenario 1 are indicated in Figure 9, with a capacity of 278–370 kW/year. By 2023–2024, the plants may fulfill 0.70 percent of the district's expected demand; Table A1 has a detailed calculation for scenario 1.

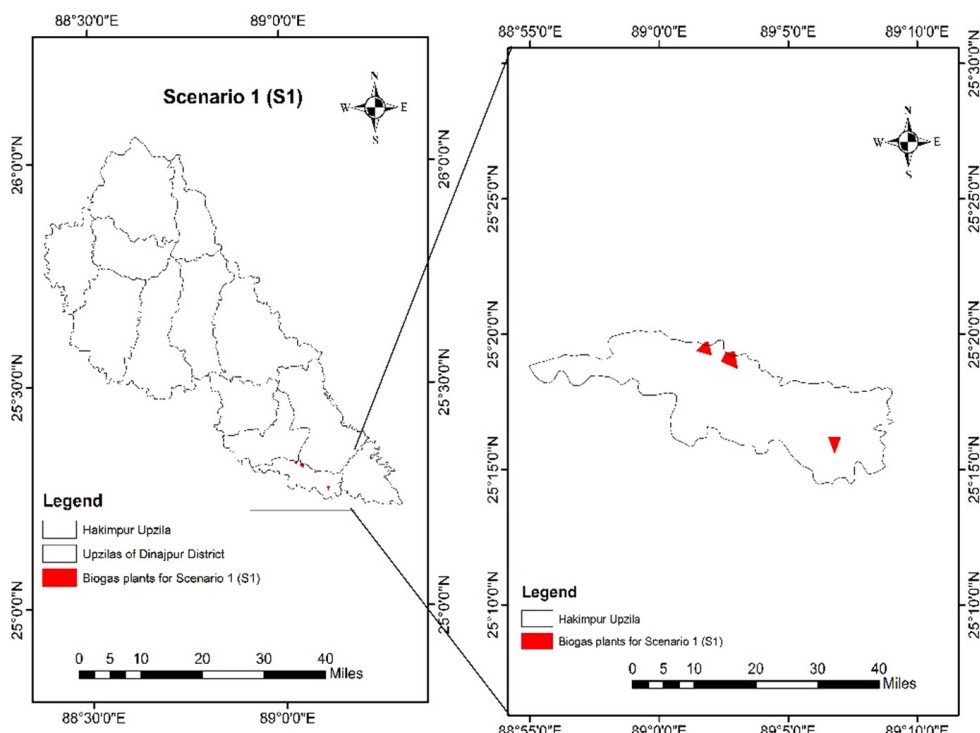

**Figure 9.** Biogas plants in Scenario 1.

### 3.6.2. Scenario 2

One biogas plant with a potential capacity of 236.11 kW/year in Dinajpur Sadar Upazila was found using intersecting analyses and capacity estimation. In 2019, it could provide 0.21 percent of the district's total energy demand. However, by 2023–2024, the plants may meet 0.18 percent of Dinajpur District's expected demand. The total residues needed for energy generation are 11,968.09 tons/year, as shown in Figure 10. Table A2 presents the potential biogas plant capacity calculation for scenario 2.

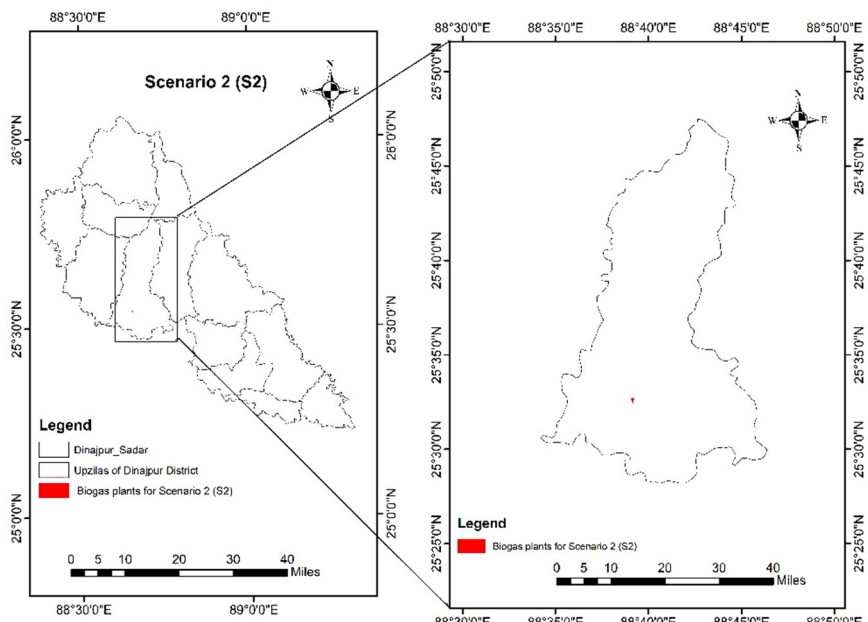

**Figure 10.** Biogas plants in Scenario 2.

### 3.6.3. Scenario 3

After combining suitability and hotspot analyses with output estimation, two potential locations for biogas plants were identified in the Dinajpur Sadar Upazila with capacities of 229.92 kW/year and 265.37 kW/year, respectively, as shown in Figure 11. The total potential electricity to be produced from these biogas plants is 495.29 kW/year, which requires total residues of 21,613.49 tons/year. Therefore, it is found that about 0.44% and 0.37% of local electricity demand in 2019 and 2023–2024 would be fulfilled by Scenario 3. Table A3 shows details of the calculations for Scenario 3.

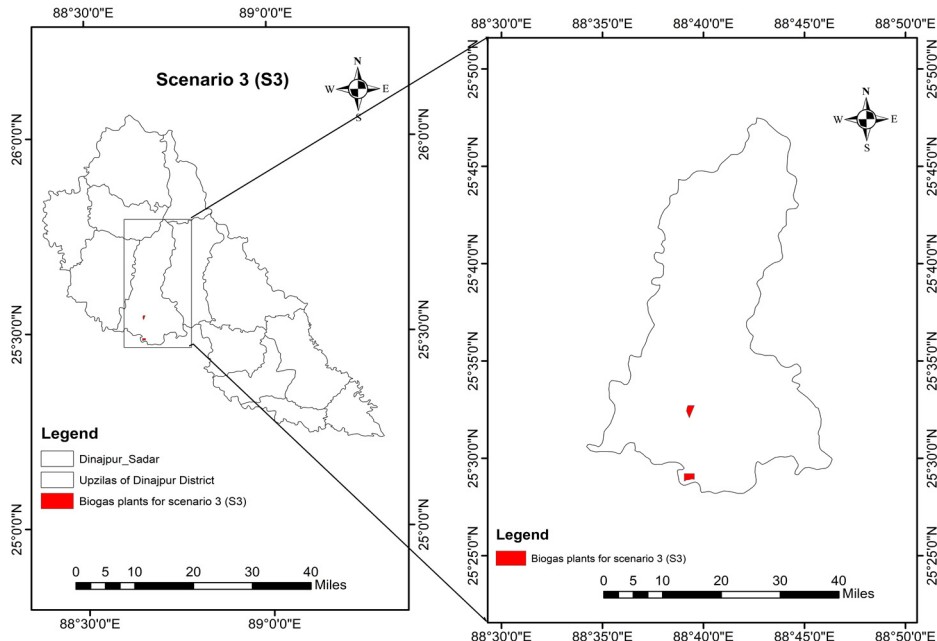

**Figure 11.** Biogas plants in Scenario 3.

### 3.6.4. Scenario 4

Eight biogas plants with a total capacity of 3047.411 kW/year were viable in the Hakimpur and Goraghat Upazilas in Dinajpur District, as displayed in Figure 12. Therefore,

four potential biogas plants in Hakimpur Upazila and the other four probable biogas plants in Goraghat Upazila can produce 1387.27 kW/year and 1660.15 kW/year, respectively, with a total residue consumption of 175,390 tons/year. Therefore, electricity from scenario 4 can meet up to 2.73% and 2.31% of total local electricity demand in 2019 and 2023–2024, respectively. Detailed calculations for Scenario 4 are shown in Table A4.

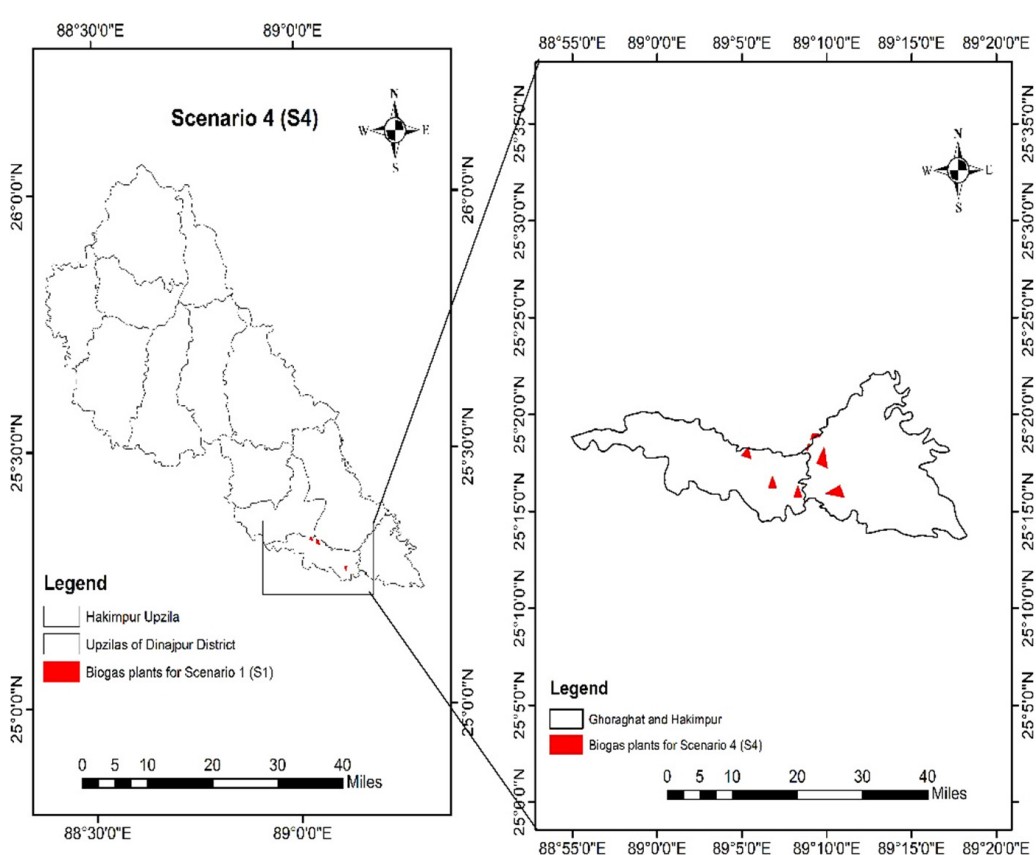

**Figure 12.** Biogas plants in Scenario 4.

### 3.6.5. Scenario 5

After intersection and capacity assessments, one biogas plant in the Dinajpur District with a capacity of 236.11 kW per year was found viable, as shown in Figure 13. Total residues of 11,968.09 tons/year are needed to produce electricity, which is capable of meeting up to 0.21% of the local demand of the district in 2019. This energy would have the capacity to account for 0.18% of the expected demand in 2023–2024. Detailed calculations for Scenario 5 are presented in Table A5.

### 3.6.6. Scenario 6

For scenario 6, six potential biogas plants in Dinajpur Sadar Upazila are feasible with a capacity of 1447.68 kW per year by applying intersection analysis of hotspots with suitable areas along with capacity assessment. The result is shown in Figure 14. The prospective biogas plants can meet 1.30% and 1.10% of the electricity demand of the district in 2019 and 2023–2024, respectively, with 62,988.73 tons of residue per year. Detailed computation on Scenario 6 is given in Table A6.

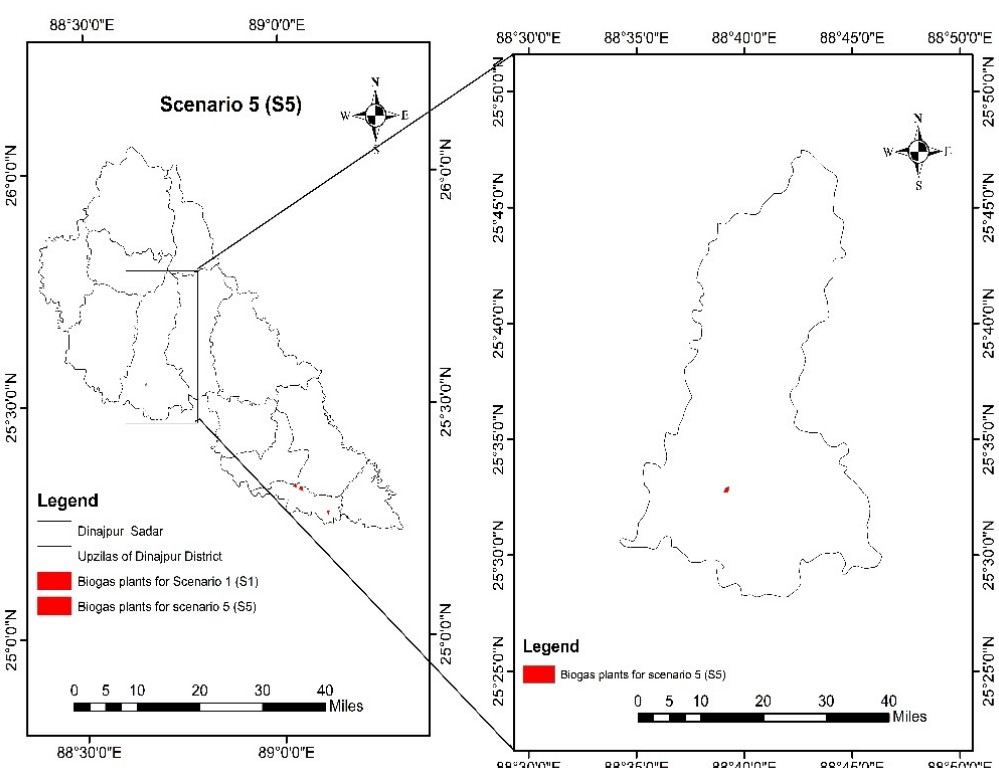

**Figure 13.** Biogas plants in Scenario 5.

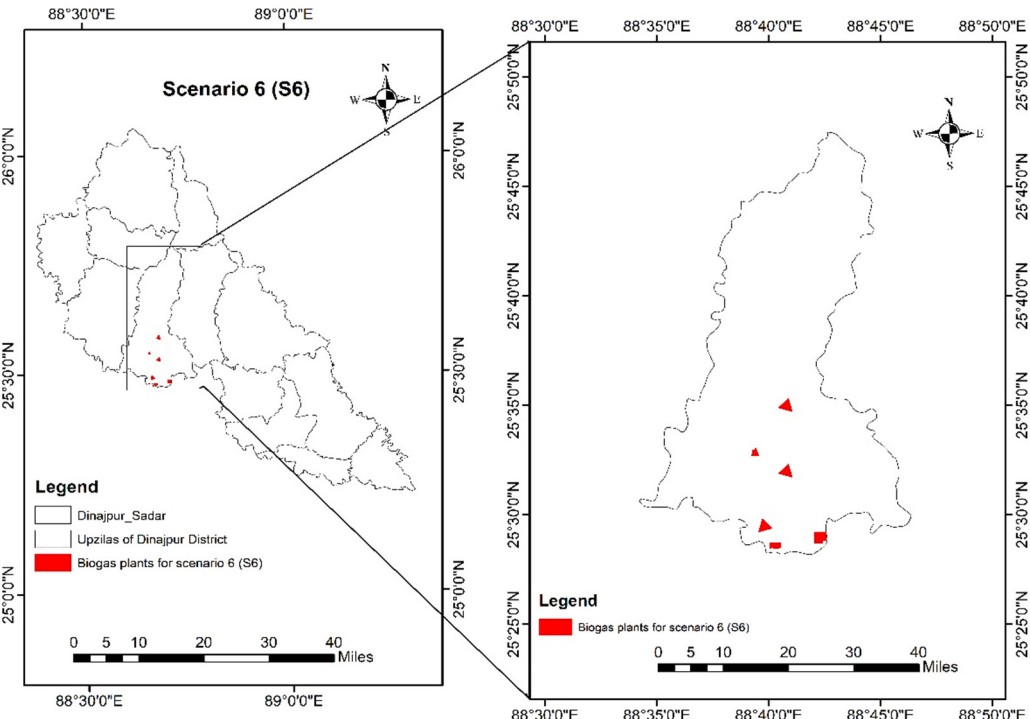

**Figure 14.** Biogas plants in Scenario 6.

Out of 21 potential biogas plants, there are 15 with outputs exceeding 250 kW/year. The results from the six scenarios demonstrate that biogas utilization partially satisfies 5.73% of the total electricity demand in the Dinajpur District. Analysis of all scenarios shows scenario 4 produced the highest amount of electricity, 3047.41 kW/year, followed by scenario 61,447.68 kW/year. However, scenario 1 amounted to 926.54 kW/year, and

scenario 2 amounted to 3495.29 kW/year, as indicated in Table 6. Conversely, Scenarios 2 and 5 would produce comparatively less energy as they have only one biogas plant each.

In scenario 1, the potential biogas of 9114.70 $m^3$ would ultimately result in 3 biogas plants with 926.54 kW per year, which can displace 8476.67 $m^3$ per year of natural gas under the business-as-usual situation (BAU) considering the 2019 energy generation in Bangladesh. Likewise, 3645.88 kg per year of furnace oil, 4739.64 L per year of diesel oil, 561.11 L per year of kerosene, and 13,307.46 kg per year of coal can be displaced. Conversely, in scenario 2, biogas produced will offset 1457.29 L per year of kerosene, furnace oil 940.19 kg per year, diesel coal 1222.25 L per year, and natural gas 2185.95 $m^3$ per year. The results are outlined in Table 8. which also highlights the bio-energy displacement of the current utilization of conventional energy sources, as listed in Figure 1 above. In scenario 1, 9114.70 $m^3$ per year of biogas would avoid 9447.38 Ton of $CO_2$ which is shown in Table 9.

**Table 8.** Comparative fossil fuel (FF) consumption assessment against potential biogas availability.

| Scenarios | Potential Biogas Production ($m^3$/Year) | Potential Electrical Energy (kW/Year) | Fossil Fuel Equivalence | | | | |
| | | | Natural Gas ($m^3$/Year) | Furnace Oil (kg/Year) | Diesel Oil (L/Year) | Kerosene Oil (L/Year) | Coal (kg/Year) |
|---|---|---|---|---|---|---|---|
| Scenario 1 | 9114.70 | 926.54 | 8476.67 | 3645.88 | 4739.64 | 5651.11 | 13,307.46 |
| Scenario 2 | 2350.48 | 236.11 | 2185.95 | 940.19 | 1222.25 | 1457.29 | 3431.70 |
| Scenario 3 | 4428.10 | 495.29 | 4118.13 | 1771.24 | 2302.61 | 2745.42 | 6465.03 |
| Scenario 4 | 12,916.25 | 1447.63 | 12,012.11 | 5166.5 | 6716.45 | 8008.07 | 18,857.73 |
| Scenario 5 | 2350.48 | 236.11 | 2185.95 | 940.19 | 1222.25 | 1457.29 | 3431.70 |
| Scenario 6 | 33,347.23 | 3047.41 | 31,012.92 | 13,338.89 | 17,340.65 | 20,675.28 | 48,686.95 |

**Table 9.** Avoided carbon dioxide emission from biogas electricity generation.

| Scenarios | Amount of Biogas from Scenario ($m^3$/Year) | Grid Emission for Bangladesh | Avoided $CO_2$ Emission (Ton $CO_{2e}$) |
|---|---|---|---|
| Scenario 1 | 9114.70 | 0.691 | 9447.38 |
| Scenario 2 | 2350.48 | 0.691 | 2436.27 |
| Scenario 3 | 4428.10 | 0.691 | 4589.72 |
| Scenario 4 | 12,916.25 | 0.691 | 13,387.69 |
| Scenario 5 | 2350.48 | 0.691 | 2436.27 |
| Scenario 6 | 33,347.23 | 0.691 | 34,564.40 |

## 4. Discussion

### 4.1. Impacts of the Theoretical Potential of Biogas Production in Dinajpur District, Bangladesh

Much literature is available on the theoretical assessment of the potential of agricultural waste in Bangladesh. Still, essential factors such as the collection factor, surplus availability factor, methane content from feedstock, etcetera, are omitted. Moreover, only single substrates, such as cattle manure or rice straw, are optimized for potential biogas assessments in the spatial analysis of past research. Moreover, the co-digestion of available feedstock is not considered for GIS suitability analysis [71]. However, this research demonstrates that the co-fermentation of agricultural and animal waste generates more biogas than fermentation.

Anaerobic co-digestion of livestock and crop residues is an environmentally acceptable method with several advantages over direct burning, including cheaper capital costs and utilizing locally accessible technology, substrate, and workforce. Another feature of this technique is the availability of yearly yield data tracking, which allows spatial analysis residue potentials to correspond with annual crop and livestock output levels to improve manure management in the Dinajpur District. Furthermore, a GIS suitability study is performed based on geographic, socioeconomic, and environmental parameters relevant

to improving the techniques used for good land management. The study highlights how potential biogas-based energy plants are determined from available agri-wastes in the proposed scenarios. The proposed methodology is pliable and can be implemented to identify agricultural waste locations and potentials in high-density areas with similar feedstocks, such as the Sirajganj, Mymensingh, Naogaon, and Munshiganj districts.

Biogas is a well-established green technology that fulfills a variety of activities related to GHG mitigation, organic waste landfilling reduction, reduction of pungent odor from open dumping, electricity generation, and circular economy promotion. Furthermore, treating animal waste with biogasification rather than open dumping or land application is a better alternative for the environment since inadequate livestock waste management causes water eutrophication and soil degradation and contributes to air pollution through greenhouse gas emissions [72,73]. It also provides an additional channel for diversifying and disseminating renewable energy systems in rural, peri-urban, and urban areas, addressing Bangladesh's National Renewable Energy Policy's objectives [74].

Furthermore, spatial distribution data of other types of manure (for example, chicken, buffalo, sheep, and goats), crop residues (for example, wheat straw, sugarcane bagasse, and rice husk), and food waste that contain more organic compounds and can be appropriated for bioenergy production while simultaneously reducing pollution. Expanding bioenergy from food and green wastes, including garden waste, could help Bangladesh diversify its renewable energy sources, especially if waste characterization and separation are prioritized. Besides, given the intended carbon-nitrogen ratios of 20.3 and 20.9 for kitchen trash and 20.9 for garden waste, compared to 10.6 to 15.8 for livestock manure, the energy potential can be significant [75].

The thermal outputs of all co-digested residue species are unidentified in the literature; until June 2021, Bangladesh imports 1160 MW from its neighbor, India [7]. As a result, in addition to energy security, the plan to cut electricity imports is required for long-term green growth. The results under single digestion show that cow dung can produce 69,951.95 MWh per year, whereas crop residue may produce 507.36 MWh annually from the accessible feedstock. The result may seem insignificant compared to the electricity provided by solar (1676 MW), wind (1370 MW), waste-to-energy (40 MW), and hydroelectric (4 MW) in 2021 [76]. For example, suppose rice straw is paired with cow dung. In that case, the leftover animal dung from cows and rice straw feedstocks could potentially create 70,459.31 MWh per year, accounting for 5.73% of the total electrical energy generated by Dinajpur District in 2019, which can offset fossil fuel-based electricity consumption by 0.02% to 0.58%, thereby reducing GHG emissions by 2436.27 to 34,564.40 tons of $CO_2$ equivalent annually.

Presently, the electricity generation capacity from renewable sources in Bangladesh is estimated at 649.51 MW [77], which equates to 1.43% of the total power capacity generation, as outlined in Figure 1. The installation of the proposed 6.389 MW biogas plants in Dinajpur District through co-digestion can cover 0.98% of renewable energy capacity. Therefore, the GIS dataset ought to be utilized by policymakers instead of disorganized manual data for evaluating and investing in agricultural waste management. If changes occur in socioeconomic situations, logistics, or public health legislation, it is also possible to design a roadmap for the transportation and collection of feedstock by decision-makers. As a result, the output maps might direct policymakers on a path of showcasing biogas plants' location, capacity, and size to achieve the goals of emission reduction of 27.56 MT $CO_2e$ (6.73%) below BAU in 2030 addressed in the Nationally Determined Contribution under the Paris Agreement of 2015 and increased renewable energy set out in the National Renewable Energy Policy of 2008.

### 4.2. A Biological Route towards Achieving Climate Neutrality

Fossil fuels have been paramount to economic growth for thousands of years, with coal, petroleum, and natural gas powering our homes and industries. The implications are immensely positive, generating vast amounts of electrical energy and powering cars,

engines, and power plants. Conversely, they have constrained our environment by causing damaging effects on ecosystems and increasing global warming [78]. The burning of fossil fuels discharges greenhouse gases such as carbon dioxide ($CO_2$), methane ($CH_4$), and nitrous oxide ($N_4O$) into the air, which are trapped in the atmosphere and lead to global warming, air pollution, water pollution, and climate change. Additionally, livestock dispense methane ($CH_4$), one of the deadliest greenhouse gases, that traps more heat in the atmosphere at a rate of more than 80 times $CO_2$ [79].

Apart from fossil fuel production, methane is also generated from the anthropogenic condition of livestock rearing, where livestock manure decomposes in large piles that are stored openly or in lagoons, typically on dairy, swine, and poultry farms [80]. In 50 years (1961–2010), global emissions from livestock rearing increased by 51%, primarily due to developing countries with vast amounts of livestock [81]. Thus, there is a critical need to mitigate GHG emissions from the livestock sector. Furthermore, constant increases in GHG emissions from various sectors affect global temperatures. For example, the increase in global mean temperature of $1.7 \pm 0.13\ °C$ above the 1900 average has led to 2015 to 2021 being the hottest years on record. Hence, countries need to decide the role of fossil fuels and animal manure management as they advance; that is, effective ways of eliminating or reducing their usage in the system must be decided. Arguably, their roles are diminishing, as they might have peaked since countries like Japan, Korea, New Zealand, the United Kingdom, and Canada have pledged net-zero emissions by 2050 [82].

A rational solution is to adopt techniques and technologies that facilitate the transition to low-carbon energy systems without compromising economic function. Biogasification through fermentation is among the renewable energy systems that provide multi-function of benefits such as improved manure management, reduced $CO_2$ emissions, decreased soil and water pollution, and encouraged circular economies while producing electrical energy. Biogas produced in rural Bangladesh is an efficacious carbon-neutral energy-based solution to improve sustainable development. Farmer adaptation of central biogas systems directly targets sustainable development goals (SDGs) number 7—affordable and clean energy, 11—sustainable cities and communities, 13—climate action, 15—life on land, and indirectly addresses goals 6—clean water and sanitation, 12—responsible consumption and production, and 17—partnerships for the purposes.

Bangladesh is a least-developed country, contributing less than 0.35% of global emissions. Nonetheless, Bangladesh recognizes that to meet the 2 degree objective, it needs to undertake mitigation in line with the IPCC conclusion that meeting 2 degrees requires global reductions of 40 to 70% in global anthropogenic greenhouse gas emissions by 2050 compared to 2010. Thus, Bangladesh's approach is driven by the long-term goal announced by its prime minister that its per capita GHG emissions must not exceed the average for developing countries [83]. In addition, the approach focuses on creating a path that maximizes the use of renewables to lower GHG emissions and ensure energy autonomy. Biogasification is a form of renewable technology that promotes a sustainable, low-carbon, technologically advanced economy.

Furthermore, biogas is a low-carbon-footprint, cost-effective technology based on the biological process of fermentation or co-fermentation of organic wastes. In this research, cow dung and rice straw are utilized as substrates in simulated computations to ascertain the possible amounts of biogas and the final valuable by-product of electrical energy. As a case study, the project promotes expanded bio-based renewable energy utilization in the Dinajpur District of Bangladesh. Additionally, it addresses several energy-based policies, including the Energy Efficiency and Conservation Master Plan up to 2030 and the Scaling Up Renewable Energy Program for Bangladesh (SREP Bangladesh). Moreover, the generation of 70,459.31 MWh per year indicates remarkable potential for direct waste-to-resource or waste-to-energy through the circular economy approach, with reduced waste for landfilling and a decreased carbon footprint by offsetting 3,324,434.54 MT of $CO_2$ emissions, providing a path to climate neutrality.

Moreover, biogas production in Dinajpur, Bangladesh, offers an opportunity to achieve an increase in renewable energy utilization of 5.73%, adding to the nation's gross inland consumption of 3%, ultimately making it an essential part of future renewable energy expansion in other rural areas of Bangladesh, entrenched on the research technique's adaptability. The results support Bangladesh's nationally determined goal under the Paris Agreement (PA) adopted by COP21 in 2015 to reduce GHG emissions by 6.73% (27.56 MT $CO_{2e}$) attributed to unconditional contribution or 15.12% (61.9 MT $CO_{2e}$) under the conditional contribution based on a baseline scenario of 169.05 MT $CO_{2e}$ in 2012 with an expected increase to 409.4 MT $CO_{2e}$ in 2030 (Ministry of Environment, Forest, and Climate Change, 2021). Biogas plays an integral role in the expected reduction, considering the energy sector accounts for 55% of the country's total emissions, with power generation equal to 23.24%. On the contrary, waste accounts for 7.55% of the 30.89 MT $CO_{2e}$ emitted.

Therefore, the estimated electricity from biogasification in Table 8 can offset 0.01% to 0.16% of total power generation and consumption emissions given the baseline scenario emissions of 2012 [83]. Under Bangladesh's 2030 scenario, where global warming is below 1.5 °C, renewable gas from biogas would increase by 5 MW [83]. However, increased investments in biogasification systems in rural communities across Bangladesh can deliver higher outputs of renewable energy from waste while directly reducing GHG emissions in the power and waste sectors. Increasing biogas for electricity facilitates the uptake of renewable gases in the energy and waste sectors while simultaneously reducing GHG emissions.

Consequently, biogas as a green technology can contribute to climate neutrality by 2050 because it decreases methane emissions from the open dumping of livestock manure, replaces conventional energy sources, produces organic fertilizer from sludge, and encourages carbon reuse through circularity. The promotion of circularity and efficient resource utilization creates a pathway to offset GHG emissions while reducing the negative carbon footprint to enhance the sustainable development of rural communities in Bangladesh.

## 5. Conclusions

The research used GIS suitability analysis with weight performance on AHP to analyze the spatial distribution of cattle manure and rice straw availability, then calculate the theoretical potential of biogas plants by employing different criteria. As a result, the appropriate location for biogas plant installation combined with improved agri-waste management could provide advantages along social, environmental, and energy paradigms and promote a circular economy by reducing agricultural waste through decentralized co-digestion of biogas production.

The results revealed that the study area, which serves as a model for rural Bangladesh, has a considerable biogas potential of 11.32 million m$^3$ per year from the combined utilization of animal waste and crop residues as substrates. The study found that 21 biogas plants can generate 70,459.309 MWh of electricity per year. All the proposed plants within the six scenarios can generate 6389.14 kW per year of electrical energy from cattle manure and rice straw, which accounted for 5.73% of total electricity generation in 2019. The government can take the initiative to implement 15 plants with a capacity of more than 250 kW as the cut value is more appealing for economic investment. Along with government help, public-private partnership initiatives are highly encouraged to finance biogas-based energy generation from available agricultural feedstocks.

Furthermore, the vast potential of biogas in rural sections of the country may be exploited for decentralized energy generation and other innovative products like heating, cooling, and lighting. After all, achieving a clean, green, and sustainable energy system from the biogasification of cattle manure and rice straw demands comprehensive technical, economic, social, and policy reforms, representing a crucial investment in Bangladesh's rural areas. Moreover, biogas production reduces challenges to health, energy, and the environment and opens a new window for the sustainable management of agricultural residues.

The social benefits hypothetically contribute to additional income for farmers and advance employment opportunities to enhance the resilience of communities to economic

and environmental shocks, thereby contributing to sustainable development in rural communities. Moreover, the flourishing development of decentralized biogas plants can replace dependency on conventional fossil fuels and create new opportunities for nutrient management. The research will help the Bangladesh government formulate a new energy policy to treat crop residues and livestock wastes by constructing biogas plants.

## 6. Limitation

This study did not consider the costs involved in every stage of livestock and agricultural waste management in the study area. The study has utilized AD for the bio-gasification of farms with >50 animals to calculate energy potential. If it is possible to encounter small farms of animals, the energy production from agricultural waste will increase. Further studies can be achieved through the financial feasibility of this project for both bio-gasification and composting facilities. To improve the proposed methodology, the author would use up-to-date and detailed GIS files of the road network for network analysis to calculate transportation costs. Therefore, the author's plan is used in this study as a reference for similar initiatives ranging from assessing environmental impact and promoting renewable energy to fulfilling the government's targets through sustainable waste management.

**Author Contributions:** Conceptualization, S.S. and H.Y.; methodology, S.S. and D.R.; formal analysis, S.S., H.Y. and D.R.; investigation, S.S. and D.R.; writing—original draft preparation, S.S. and D.R.; writing—review and editing, S.S., D.R. and H.Y.; visualization, S.S. and D.R.; supervision, H.Y. All authors have read and agreed to the published version of the manuscript.

**Funding:** This research received no external funding.

**Institutional Review Board Statement:** Not applicable.

**Informed Consent Statement:** Not applicable.

**Data Availability Statement:** All data are reported in this work.

**Acknowledgments:** The authors of this research are grateful for the help of Professor Mizunoya Takeshi who gave crucial advice during the research process. We also thank the University of Tsukuba, the Graduate School of Science, Technology, and Information Science, the Department of Life and Environmental Sciences, the Department of Livestock Services of the Ministry of Fisheries and Livestock, Bangladesh, and the Bangladesh Bureau of Statistics for the provision of essential data included in the study.

**Conflicts of Interest:** The authors declare no conflict of interest.

## Appendix A

**Table A1.** Calculation for Scenario 1.

| FID | Source ID | Grid Code | Gi_Bin | Available Livestock Residue (T/year) | Available Crop Residue (T/year) | Area | Energy (kWh/Year) | Capacity (kW) |
|---|---|---|---|---|---|---|---|---|
| 0 | 69 | 3 | 1 | 496.89 | 12,783.16 | Hakimpur | 5773.75 | |
| 1 | 70 | 3 | 1 | 496.89 | 12,783.16 | Hakimpur | 5773.75 | 926.54 |
| 2 | 78 | 3 | 1 | 472.88 | 19,174.72 | Hakimpur | 8403.16 | |

**Table A2.** Calculation for Scenario 2.

| FID | Source ID | Grid Code | Gi_Bin | Available Livestock Residue (T/year) | Available Crop Residue (T/year) | Area | Energy (kWh/Year) | Capacity (kW) |
|---|---|---|---|---|---|---|---|---|
| 0 | 33 | 3 | 2 | 355.97 | 11,612.12 | Dinajpur Sadar | 5138.66 | 236.11 |

**Table A3.** Calculation for Scenario 3.

| FID | Source ID | Grid Code | Gi_Bin | Available Livestock Residue (T/Year) | Available Crop Residue (T/Year) | Area | Energy (kWh/Year) | Capacity (kW) |
|---|---|---|---|---|---|---|---|---|
| 0 | 33 | 3 | 3 | 471.84 | 9825.67 | Dinajpur Sadar | 4501.68 | 495.29 |
| 1 | 43 | 3 | 3 | 597.11 | 10,718.87 | Dinajpur Sadar | 4985.04 | |

**Table A4.** Calculation for Scenario 4.

| FID | Source ID | Grid Code | Gi_Bin | Available Livestock Residue (T/year) | Available Crop Residue (T/year) | Area | Energy (kWh/Year) | Capacity (kW) |
|---|---|---|---|---|---|---|---|---|
| 0 | 71 | 4 | 1 | 472.89 | 19,174.72 | Hakimpur | 8381.88 | |
| 1 | 73 | 4 | 1 | 472.89 | 19,174.72 | Hakimpur | 8381.88 | |
| 2 | 74 | 4 | 1 | 472.89 | 19,174.72 | Hakimpur | 8381.88 | |
| 3 | 74 | 4 | 1 | 496.89 | 12,783.16 | Hakimpur | 5751.39 | 3047.41 |
| 4 | 75 | 4 | 1 | 186.86 | 25,604.93 | Goraghat | 10,792.58 | |
| 5 | 76 | 4 | 1 | 186.86 | 25,604.93 | Goraghat | 10,792.58 | |
| 6 | 79 | 4 | 1 | 186.86 | 25,604.93 | Goraghat | 10,792.58 | |
| 7 | 80 | 4 | 1 | 186.86 | 25,604.93 | Goraghat | 10,800.98 | |

**Table A5.** Calculation for Scenario 5.

| FID | Source ID | Grid Code | Gi_Bin | Available Livestock Residue (T/year) | Available Crop Residue (T/year) | Area | Energy (kWh/year) | Capacity (kW) |
|---|---|---|---|---|---|---|---|---|
| 0 | 33 | 3 | 2 | 355.97 | 11,612.12 | Dinajpur Sadar | 5138.66 | 236.11 |

**Table A6.** Calculation for Scenario 6.

| FID | Source ID | Grid Code | Gi_Bin | Available Livestock Residue (T/Year) | Available Crop Residue (T/Year) | Area | Energy (kWh/Year) | Capacity (kW) |
|---|---|---|---|---|---|---|---|---|
| 0 | 29 | 4 | 3 | 406.59 | 8039.18 | Dinajpur Sadar | 3701.67 | |
| 1 | 31 | 4 | 3 | 471.84 | 9825.67 | Dinajpur Sadar | 4501.68 | |
| 2 | 33 | 4 | 3 | 471.84 | 9825.67 | Dinajpur Sadar | 4501.68 | 1447.68 |
| 3 | 40 | 4 | 3 | 597.11 | 10,718.87 | Dinajpur Sadar | 4985.04 | |
| 4 | 44 | 4 | 3 | 597.12 | 10,718.87 | Dinajpur Sadar | 4985.04 | |
| 5 | 45 | 4 | 3 | 597.11 | 10,718.87 | Dinajpur Sadar | 4985.04 | |

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
