# Peer review of "Green Energy Optimization in Dinajpur, Bangladesh: A Path to Net Neutrality"

_sustainability, doi:10.3390/su15021336_

Round 1

Reviewer 1 Report

Overall, it is a good manuscript, but several things are needed to improve. 1) Introduction described too much about the importance of renewable energy, but should offer more informantion about why the assessment of substrate availability and location is important. 2) The section 1.1 "role of anaerobic fermentation in the biogas process is not necessarily needed", too much description about anaerobic digestion. 3) The reference used in Table 1. and Table 2. are reqired more recently. 4) Have the authors considered about the conversion from cattle manure and rice residue to biogas and biogas to electricity? 5) Any comparable research can be support author's hypothesis?

Author Response

Thank you very much for kindly taking the time to revise our manuscript. Please find the response to your comments/suggestions in the attached file.

Sincerely;

Helmut Yabar

Reviewer 2 Report

The title of the manuscript is not suitable. Revise the title using novel results or methodology.

Most of the Abstract belongs to the materials and methods. I suggest to short this up to 1-2 sentences. Also, add the importance and objectives of the study briefly. 

L11, Don't use citations in the abstract.

Revise the keywords, as these are not suitable.

Where is the novelty of the study? Also, add the hypothesis of the study.

L328, Don't use paragraphs with abbreviations. Revise the whole manuscript to fix this issue. 

Future perspectives and important is very important for this study. 

Author Response

Thank you very much for taking the time to kindly revise our manuscript. Please find the response to your comments/suggestions in the attached file.

Sincerely;

Helmut Yabar

Round 2

Reviewer 2 Report

I accept all changes to the manuscript and no further changes require